# A Novel Improved Bat Algorithm Based on Hybrid Parallel and Compact for Balancing an Energy Consumption Problem

**Trong-The Nguyen [1,2], Jeng-Shyang Pan [1,3,4,*] and Thi-Kien Dao [1,2,*]**

[1] Fujian Provincial Key Laboratory of Big Data Mining and Applications, Fujian University of Technology, Fuzhou 350014, China; vnthe@hpu.edu.vn

[2] Department of Information Technology, University of Manage and Technology, Haiphong 180000, Vietnam

[3] College of Computer Science and Engineering, Shandong University of Science and Technology, Qingdao 266510, China

[4] College of Informatics, Chaoyang University of Science and Technology, Taichung 413, Taiwan

[*] Correspondence: jengshyangpan@gmail.com (J.-S.P.); jvnkien@gmail.com (T.-K.D.)

**Abstract:** This paper proposes an improved Bat algorithm based on hybridizing a parallel and compact method (namely pcBA) for a class of saving variables in optimization problems. The parallel enhances diversity solutions for exploring in space search and sharing computation load. Nevertheless, the compact saves stored variables for computation in the optimization approaches. In the experimental section, the selected benchmark functions, and the energy balance problem in Wireless sensor networks (WSN) are used to evaluate the performance of the proposed method. Results compared with the other methods in the literature demonstrate that the proposed algorithm achieves a practical method of reducing the number of stored memory variables, and the running time consumption.

**Keywords:** improved bat algorithm; optimization deployment problems; probabilistic model; wireless sensor network

## 1. Introduction

Metaheuristic algorithms are one of the most main potential tools for solving complex optimization problems. Metaheuristic algorithms have been applied successfully to optimization problems in the fields of engineering, biology, and finance [1–3]. The Bats Algorithm (BA) is a novel meta-heuristic search algorithm [4], which simulates the behavior of the bats species for searching prey. Preliminary studies show that it is very promising and could outperform existing algorithms [5–7]. BA utilizes a population of bats to represent candidate solutions in a search space and optimizes the problem by iteration to move these agents to the best solutions. The general steps of this algorithm are described in the next section. In addition, the original BA can solve problems with continuous search space, while several versions of the algorithm are also proposed in the literature to solve problems with continuous and discrete search spaces. An Evolved bat algorithm (EBA) is used for numerical optimization and the economic load dispatch problem [8,9]. A hybrid between BA and Artificial bee colony (ABC) is used for solving numerical optimization problems [10]. In addition to continuous BAs, several discrete BAs have also been proposed in the literature. A binary BA (BBA) was proposed to solve the feature selection problem [11], where its solution is restricted to be a vector of binary positions using a sigmoid function. A similar BBA algorithm with a multi v-shaped version of the transfer function was adopted to solve large scale 0–1 knapsack problems [12]. Another version of BA was discretized for job shop scheduling problems [13]. However, BA has not considered the saving variable memory, and it has not

been given the selected parameters of the algorithm based on the objective function, so the optimal performance will not be very effective in removing the hot spot problem.

Moreover, the rapid growth in the field of integrated circuits (IC) and Information technology (IT) has led to the development of cheap and compact size sensor nodes of Wireless sensor work (WSN) [14]. WSN is a promising and emerging technology, and it is an essential part of the Internet of Things (IoT) infrastructure for collecting relevant information in the target environment. WSN is composed of a set of a vast number of sensor nodes that are operated in an ad-hoc fashion to observe and interact with the physical world. WSN has been widely applied in a variety of fields of industry, traffic control, healthcare, and home automation [15,16]. However, the sensor nodes are limited in computing capability and storage capacity of the computing unit, in communicating the range and radio quality of the communication unit, in sensing the coverage and accuracy of the detecting unit, and in the available energy of power units [17,18]. Because of the limited memory and the constrained power, fully functional WSNs must be maintained and kept stable by the sound design employed network.

The clustering method in WSN is one of the most outstanding energy efficient ways of saving the energy network. The clusters are generated by arranging the sensor nodes into groups. A cluster has Node members (NM), and a Cluster head (CH) that are selected among NM. Clustering provides various advantages like energy efficiency, lifetime, scalability, and less delay. However, clustering can lead to a hot spot problem. Further, the unequal clustering technique is utilized for load balancing among CHs to prevent the network from developing a hot spot issue [19]. In uneven clustering, the cluster size varies proportionally to the distance to the base station (BS). Figure 1 shows the architecture of solving the unequal clustering in WSN. The size of the clusters would be reduced regardless of whether it is closer to BS.

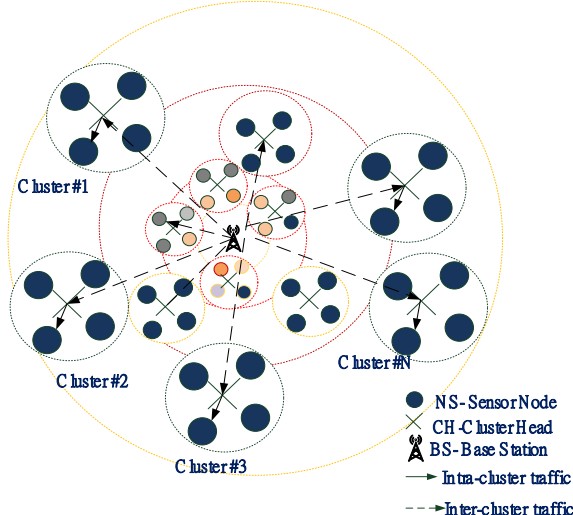

**Figure 1.** The architecture of uneven clustering in WSN.

In contrast, the cluster size would be increased if the distance between the BS and CH is great. The cluster size is directly proportional to the length of CHs from BS. Unequal Clustering permits all CHs to pay the same amount of energy so that the CHs near BS spend equal energy to the CHs farther from BS. So, uneven grouping eliminates the hot spot problem by balancing the load efficiently. Due to the dense deployment and unattended nature of WSN, it is hard to recharge node batteries. The efficient energy and maximize network lifetime is a primary design goal in deployed WSN.

Several traditional approaches had dealt with unequal clustering, e.g., the probabilistic, deterministic, and heuristic approaches [20]. However, if the scale network is vast, traditional methods would require a long time for computation and the accuracy would decrease. The metaheuristic algorithm is the preferred method that can apply to address the hotspot problem adequately [21].

Challenges to the optimization applications have also arisen somehow from the limited hardware resource due to the cost, storage or space size. A compact optimization algorithm would decrease the variables of candidating solutions, but it still obtained good results [22].

In this paper, we extend our previous conference papers [7,23] by hybridizing the parallel with the compact techniques for the numerical optimization problems and balancing the energy consumption problem in WSN. The logic behind extending works includes a hybrid parallel and compact, an added weight to control the probability of sampling for the perturbation vector, and selected parameters for the balancing the energy consumption problem of WSNs. The parallel processing considers the diversity of communication strategies. The compact technique considers improving the built probability model. The poor sampling individuals in the subgroups have been replaced with the better sampling competition agents according to the fitness evaluation.

The rest of the paper is organized as follows: Section 2 provides a brief review of BA and a statement hotspot problem in WSNs. Section 3 presents analysis and a design for hybrid parallel and compact BA (pcBA). The simulation test and results are discussed in Section 4. A solution to the balanced energy consumption problem in WSN is figured out in Section 5. Section 6 summarizes the conclusion.

## 2. Related Work

### 2.1. Bat-Inspired Algorithm

The inspiration for the Bats algorithm (BA) [4] was drawn from the echolocation of the species called the microbats for searching prey. The update solutions of BA were constructed based on three primary characteristics, which included echolocation, frequency, and loudness. The echolocation of Bats is used to locate the prey. The frequency is used to send out the variable wavelength. The loudness is used to search for the victim. Solutions of BA are adjusted according to evaluate objective function by using certain parameters, e.g., frequencies, loudness, and the pulse emission rates of the bats. Formulas for updating the positions and velocities of BA in d-dimensional search space are as follows.

$$f_i = f_{min} + (f_{max} - f_{min}) \times \beta \tag{1}$$

where $f_i$ is the frequency for adjusting velocity change; $f_{min}$ and $f_{max}$ are the minimum and maximum frequency of the bats emitting the pulse; $\beta$ is a generated vector randomly based on distributed Gaussian $\in[0, 1]$. A frequency assigned initially for each bat in a uniform range $\in[f_{min}, f_{max}]$. BA updates the vectors of the bat's location and velocity $x$, and $v$ in the d-dimensional search space.

$$v_i^t = v_i^{t-1} + \left(x_i^{t-1} - x_{best}\right) \times f_i, \tag{2}$$

$$x_i^t = x_i^{t-1} + v_i^t, \tag{3}$$

where $t$ is the current iteration, $x_{best}$ is the global best location. Generating a new location of the bats in exploiting the phase strategy is formulated as.

$$x_i = x_i + \varepsilon \times A^t, \tag{4}$$

where $\varepsilon$ is a random variable in the range $\in [-1, 1]$, and it indicates the weight for the loudness of the bats at the current generation. The loudness of bats $A$ is defined as.

$$A_i^{t+1} = \alpha \times A_i^t, \tag{5}$$

where $\alpha$ is a variable constant. The symbol denotes the rate of the pulse emission $r$ and $\in[0, 1]$. The pulse emission rate is calculated as.

$$r_i^{t+1} = r_i^0 \times \left[1 - e^{-\gamma \times t}\right], \tag{6}$$

where $\gamma$ is a variable constant. This rate $r$ is considered in the process to switch the global and local search. If a random number is greater than $r$, a local search with a random walk is triggered.

### 2.2. Statement Problem in WSNs

Balancing energy consumption for a hierarchy WSN is done using a practical clustering approach for the hotspot issue in WSN [19]. The wireless radio transceivers in WSN depend on the various parameters, e.g., distance, energy consumption, the distance between the transmitters. Its receiver obeyed the attenuated transceiving power that decreased exponentially with the increasing distance. Dissipated energy of CH includes the power of aggregating the sensed information, transmitting the aggregated signal to the base station BS, and receiving signals from the nodes. If a data message is a number of l bits, the energy consumption of a node would be formulated as follows.

$$E_t(i)_{consumed} = \begin{cases} (m_n - 1) \times l \times E_{ele} + m_n \times l \times E_{DA} + l \times E_{ele} + l \times E_{mp} \times d_{i\ toBS}^4, & i \in CH \\ l \times E_{ele} + l \times E_{fs} \times d_{n\ to\ CH}^2, & \text{otherwiswe; it means } i \in non - CH \end{cases} \tag{7}$$

where $E_t(i)_{consumed}$ is the consumed energy of $CH_i$ ($i \in CH$); $n$ is member nodes in a round $t$. The distance from the CH to the *BS* is set to $d_{i\ to\ BS}$. The distance of the member node to CH is set to $d_{n\ to\ CH}$. The sensor nodes are connected to CH are set to $m_n$. Member nodes only need to transfer data to the CH once during a round. Presumably, the distance to the CH is small, so the energy dissipation follows the Friis free space model ($d^2$ for lost energy). Since BS is the far distance from the nodes, presumably the consumed energy follows the multi-path model ($d^4$ for lost power). Parameters of consumed energy for communication include the initial power for the nodes $E_j$, the radio electronics dissipates for receiving and transmitting units $E_{ele}$, the amplifier energy $E_{mp}$ and $E_{fs}$, and the energy of data aggregation $E_{DA}$ [24].

The average dissipated energy for round $t$ is calculated as:

$$\mu(E_{consumed}) = \left(\sum_{i \in N} E_t(i)_{consumed}\right)/N \tag{8}$$

The remaining power of cluster nodes for the round is defined as:

$$E_{t+1}(i)_{res.} = E_t(i)_{res.} - E_t(i)_{consumed} \tag{9}$$

where $E_{t+1}(i)_{res}$ is residual energy for the round ($t + 1$). The average residual power for round $t$ is calculated as:

$$\mu(E_{res.}) = \sum_{n \in N} \frac{E_t(n)_{res.}}{N} \tag{10}$$

The obtained residual energy for standard deviation is defined as:

$$\delta(E_{res.}) = SQRT(\sum_{n \in N} \frac{(\mu(E_{res.}) - E(i)_{res.})^2}{N}) \tag{11}$$

Average energy consumption of the network for around the denoted $\mu(E_{consumed})$ in Equation (8) is minimized to save energy in the cluster nodes. To balance the energy load of nodes, we use Equations (9)–(11) for minimizing the standard deviation of residual energy $\delta(E_{res})$ in WSN. The average

residual energy and number of received items are optimized to prolong the sensor network's lifetime by applying a clustering evaluation model for measuring the performance.

## 3. Methodology of Parallelized Compact BA (pcBA)

This section presents an improved BA based on hybridizing the parallel with the compact for the numerical optimization problems and balancing the energy consumption problem in WSN. The improvement BA is an extension of our previous works [7,23] that considered two techniques of parallel and compact. The parallel with a communication strategy is significant for computations that exchange information with other groups, share the computation load, and enhance the diversity of individuals [25,26]. The compact technique can offer an effective way of using a saving variable memory. An efficient compromise is used to present solutions of search space for the advantages of population-based algorithms without requirements of storing actual populations of solutions. The compact algorithm simulates the behavior of population-based algorithms by employing the replacement of a community of solutions with its probabilistic representation.

### 3.1. Parallelized Bats Algorithm

The parallel method with communication strategies in the metaheuristic algorithm is proven to have faster convergence and more accuracy than the original algorithm [25–27]. The processing parallel plays a significant role in computational optimization, which is a carried computational form that operates both in the same direction and simultaneously [23,27]. To build a parallel structure, several subpopulations are created by dividing the population in ways that could evolve separately over iterations, and the best agents are selected to continually search the next generation according to the measured fitness. The communication scheme in parallel processing exchanges their properties among groups, e.g., moving, copying, immigrating, or replacing randomly. A promising region would swap with weak areas within the solution space, and exploration of a promising area is carried out in the searching space.

---

**Scheme 1** A pseudo-code of a parallel with communication strategies

---

**Input**: The subgroups $G_i$, ... $G_m$, $i = 1, 2..., m < N_p$ the population size
**Output**: Promising regions in subgroups $G$ after communicating.
   1:　**if** $m > 2$ **then**
   2:　　**if** *random* $\leq \omega$　**then** // Stategy1- neighboring groups
   3:　　　**for** $i = 1$ to $m$ **do**
   4:　　　replace the *worst*($G_i$) with *best* random from ($G_i$, ... $G_m$)
   5:　　　**endfor**
   6:　　**else**　　　// Stategy2- the best to all
   7:　　　**for** $i = 1$ to $m$ **do**
   8:　　　replace the *worst* ($G_i$) with *best* ($G_i$, ... $G_m$)
   9:　　　**endfor**
 10:　　**endif**
 11:　**else**　　　// Stategy3- a pair swapping
 12:　　replace *worsts*($G_1$) with *best*($G_2$)
 13:　　replace *worsts*($G_2$) with *best*($G_1$)
 14:　**endif**

---

The communication strategies suggested include the best to all, neighboring groups, a pair swapping, etc. The procedure with the best to all has the most excellent agents among all subpopulations migrate to every group, mutate them by replacing the worst bats in each of these groups and update them after the period exchanging time of running. The strategy with the neighboring groups is to move the best bat of one group to its surrounding groups, then replace some poorer bats after the

period of running. The strategy with a pair swapping is a pair of two subgroups where the most exceptional bat of this subgroup replaces the worst bat in the other subset and vice-versa. The drawback of communication in those algorithms was fixed for picking one of the communication strategies. The effectiveness of these strategies therefore could not be taken advantage of. In this section, a new communication scheme is proposed to overcome this drawback. In the draft scheme, the exchanging procedures are combined rationally based on a switching parameter as the weight control. In the experimental section, a weight control $\omega$ is set to 0.7 to 0.5. This scheme dynamically enhances the communication strategies. Scheme 1 shows a pseudo code of communication scheme, namely *Communication (subpop).*

### 3.2. Compacted Bat Algorithm

The estimated distribution algorithm (EDA) process uses a probabilistic representation to get fewer stored variables, instead all the population of solutions was stored in the metaheuristic algorithms while still getting the same obtained result of optimization [28]. The compact method uses the principle of EDA to simulate the operations of the metaheuristic algorithm [7,29]. A probabilistic model was used to represent the operations of the population-based algorithm in a compact one. In this case, a real population considered as a virtual population in the compact algorithm. The virtual community is configured by probability density functions (PDFs) [30] based on EDA. Not all of the population of a solution was stored in memory, but it generated a few new solution candidates based on probability distribution stored in the memory. An attracted new candidate solution is being iteratively biased toward a promising area of an optimal solution. The likelihood of a population of individuals in an algorithm represents the probability vector of each component learned from previous generations. The structure of this vector was called the Perturbation Vector (PV) [29]. These principles were applied to the improvement of memory saving variable for compact BA.

Different from the population-based algorithms such as BA, the compact technique considered population as "virtual community" by expressing the encoded data structure of a probabilistic vector. A real-valued prototype vector represents the probability of each component being described in a candidate solution. The specified probability for each element in new candidate solutions was maintained in the optimum process. The optimization processing objective of the compact algorithm is to simulate the behavior of Bats of BA, but it was used with a much smaller stored variable memory. PV generates a candidate solution probabilistically from the vector. Competing for components toward the better solutions is reflected in the updated probability vector. The created trial solutions stayed to be allocated in boundary constraints. PV is a matrix for specifying the two parameters of mean $\mu$ and standard deviation $\sigma$ values in the PDF of each design variable. It can be defined as: $PV^t = \left[\mu^t, \sigma^t\right]$, where $t$ is the time steps.

A truncated Gaussian (PDF) for $\mu$ and $\sigma$ values are within the interval of $(-1, +1)$. The PDF normalizes the amplitude of area equal to 1. We use $PV(\mu_i, \sigma_i)$ to generate the candidate solution, where $x_i$ in the compact method. The solution is corresponding to the virtual bats based on the associated Gaussian of $\mu$ and $\sigma$ as the following expressed PV.

$$P_i(x) = \frac{\exp\left(\frac{-(x-\mu_i)^2}{2\sigma_i^2}\right) \times \sqrt{\frac{2}{\pi}}}{\sigma_i\left(\text{erf}\left(\frac{\mu_i+1}{\sqrt{2\sigma_i}}\right) - \text{erf}\left(\frac{\mu_i-1}{\sqrt{2\sigma_i}}\right)\right)} \tag{12}$$

where $P_i(x)$ is the probability distribution of $PV$ that is associated to the $\mu$ and $\delta$ in a truncated Gaussian PDF. This is the corresponding value of the PDF to variable $x_i$. The error function indicated as *erf* is found in reference [31]. PDF could have found to be corresponding to the Cumulative Distribution Function (CDF) by constructing Chebyshev polynomials [32]. The arranged codomain of CDF is from 0 to 1. The described *CDF* is a real-valued random variable $X$ in a value given distribution at $\leq x_i$. The value of the newly calculated candidate $x_i$ is a value of its inversed CDF.

### 3.3. Parallel Compact Bat Algorithm

This subsection presents an implementation of a hybrid of the parallel and compact methods for BA. Construct parallel, whole population split into several subpopulations, and the communication scheme are all triggers among the subpopulations. The subpopulations run in parallel and evolve independently based on BA optimization. The communicating subgroup is carried out, e.g., the most excellent bats among the subgroups immigrated to another subset and were replaced with the weakest bats according to a measured fitness, and the subgroups were updated over the period. In the phase of deployment compact, we figure out the pact to the subgroups based on probability vectors through the competition. The hybrid of the parallel and compact process is described through the illustrations in Schemes 2–7.

We extended a couple of improvements for the perturbation vector through the process of sampling and updating PV. New candidates are generated by learning and sampling from explicit probabilistic models that forward to promising solutions in search space. To control the probability of sampling of $\mu_i$ in PDF, a parameter $\tau$ as a weight is suggested from between left $[-1, \ \mu_i]$ and right $[\mu_i, 1]$. Thus, PV can be computed into two sides of the left and right as follows:

$$
\left|
\begin{array}{l}
L_i(x) = \dfrac{-\sqrt{\frac{2}{\pi}}}{\sigma_i \times \left(\text{erf}\left(\frac{\mu_i+1}{\sqrt{2}\sigma_i}\right)\right)} \times \exp\left(-\frac{(x-\mu_i)^2}{2\sigma_i^2}\right) \quad \text{for} -1 \leq x \leq \ \mu_i \\[4mm]
R_i(x) = \dfrac{-\sqrt{\frac{2}{\pi}}}{\sigma_i \times \left(\text{erf}\left(\frac{\mu_i-1}{\sqrt{2}\sigma_i}\right)\right)} \times \exp\left(-\frac{(x-\mu_i)^2}{2\sigma_i^2}\right) \quad \text{for } \mu_i \leq x \leq \ 1
\end{array}
\right.
\tag{13}
$$

The new sampling extended approach for PDF is for generating new candidates of the group depicted in Scheme 2.

It means the PV scheme will generate the agents of $x$ randomly.

---

**Scheme 2** Perturbation Vector (PV) for generating new solutions

---

**Input**: parameter $\mu$, $\sigma$ of probability vector, dimension $d$, and $\tau$
**Output**: A new candidate $x$
　1: **for** $i = \ 1$ to $d$ **do**
　2:　　Generated $r \in [0,1]$ randomly in uniform distribution
　3:　**if** $r < \tau$ then
　4:　　Generating $x_i \ \in [1,0]$ via $L_i(x)$ of Equation (13)
　5:　**else**
　6:　　Generating $x_i \ \in [1,0]$ via $R_i(x)$ of Equation (13)
　7:　**end if**
　8: **end for**

---

Scheme 3 shows the initializing PV scheme as the pseudo code of compact BA (cBA). The best bat $x_{best}$ is computed based on the learning scheme of a sampling trial bat. If the temporary new solutions are better according to the evaluated fitness, $x$ is then updated. $k$ is a large constant, e.g., $k$ is set to 10.

---

**Scheme 3** Initialization of cBA

---

　1:　Initialization of PV$(\mu, \sigma)$
　　　**for** $i = 1{:}n$ **do**
　　　　　$\mu_i^t = 0;$
　　　　　$\sigma_i^t = k;$
　　　**end for**
　2:　Initializing Bats location $x$ via PV
　3:　Initializing $x_{best}$ with the best location value: $x_{best} = \arg \min f[x]$.

---

Two design variables of winner and loser compete together to find out who is the better one according to the evaluated fitness value. The winner is moved toward a promising area in a searching space based on the comparison between two design variables for bats of the subgroup. A newly selected candidate is assigned to evaluate the given objective function. Determining a winning solution is employed based on the comparison of the chosen candidate agents. Scheme 4 displays the competing scheme for the winner/loser.

---

**Scheme 4** Compete for *winner* and *loser*

---

**Input**: The objective function $f$ and solutions $x_a$, $x_b$
**Output**: winner or loser
  1:  **if** $f(x_a) < f(x_b)$ **then**
  2:     *winner* assigned to $x_a$
  3:     *loser* assigned to $x_b$
  4:  **else**
  5:     *winner* assigned to $x_b$
  6:     *loser* assigned to $x_a$
  7:  **end if**

---

Moreover, the elements $\mu_i^{t+1}$ and $\sigma^{t+1}$ of the updating PV for the new solution for the winner and loser are expressed over the differential iterations. A typical parameter called virtual population $Np$ is not strict variable corresponding to the population size variable as in a population-based algorithm.

$$\mu_i^{t+1} = \mu_i^t + \frac{1}{N_p}(winner_i - loser_i) \tag{14}$$

where $t$ is current iteration, and $i = 1, 2, \ldots N_p$. Regarding $\sigma$ values, the update rule of each element is given:

$$\sigma_i^{t+1} = \sqrt{(\sigma_i^t)^2 + \left(\mu_i^t\right)^2 - \left(\mu_i^{t+1}\right)^2 + \frac{1}{N_p}\left(winner_i^2 - loser_i^2\right)} \tag{15}$$

Another improvement for updating PV, the values of $\mu_i^{t+1}$ and $\sigma_i^{t+1}$ are modified with a control parameter for expressing the maximum value of perturbation. A parameter $\vartheta$ is added as a weight control of the expressed the perturbations maximum value.

$$\mu_i = \mu_i + \beta_i \times \vartheta \tag{16}$$

$$\sigma_i = \sqrt{\sigma_i^2 + \alpha_i \times \vartheta} \tag{17}$$

where $\beta_i$ and $\alpha_i$ are random number distributed in [–1, 1], and distributed in [0, 1], respectively. Evaluate fitness function with selected location $x$ compared with $x_{best}$ to obtain a winner for the next generation. Current location $x$ maintained in following steps of the scheme. Scheme 5 shows the updating PV scheme.

---

**Scheme 5** Updating PV for new candidates

---

  1: **for** $i = 1$ to $d$ **do**
  2:   $\mu_{backup} = \mu_i^t$
  3:   $\mu_i^{t+1} = \mu_i^t + \frac{1}{N_p}(winner_i - loser_i)$
  4:   $\sigma_i^{t+1} = \text{SQRT}\left(\max\left(0, \ (\sigma_i^t)^2 + \left(\mu_{backup}^t\right)^2 - \left(\mu_i^{t+1}\right)^2 + \frac{1}{N_p}\left(winner_i^2 - loser_i^2\right)\right)\right)$
  5:   Improving for $\mu_i^{t+1}$ and $\sigma^{t+1}$ via Equations (16) and (17) with $\tau$ set to 0.01
  6:  **end for**

---

Scheme 6 indicates the pseudo-code of the steps for the compact BA. It simulated the behavior of population-based algorithms by sampling the probabilistic model. A virtual population is encoded in its probabilistic representation.

---

**Scheme 6** The compact BA, (namely *cBA)*

---

**Input**: The objective function $f$, $t = 0$ and the swarm
**Output**: The best solution $x_{gbest}$, $F_{min}$
　　1:　Initialization phase according to Scheme 3
　　2:　　**while** stop criteria are not met **do**
　　3:　　　Generating $x$ by PV, via Scheme 2
　　4:　　　Update Bats via Equations (1) to (3)
　　5:　　　Select best by Compete scheme via Scheme 4
　　6:　　　[*winner, loser*] = *compete(x, newx)*;
　　7:　　　$F_{new}$=$f$ (*newx*);
　　8:　　　Update PV scheme $\mu^{t+1}$, $\sigma^{t+1}$, via Scheme 5
　　9:　　　Global best update
　　11:　　　[*winner, loser*] = *complete(newx, $x_{best}$)*;
　　12:　　　**if** ($F_{new}$<$F_{min}$)
　　13:　　　　$x_{best}$ = *winner*; $F_{min}$=$F_{new}$;
　　14:　　　**end if**
　　15:　　　$t = t + 1$;
　　16:　　**end while**

---

The steps of the parallel compact BA algorithm are described as follows. For the first step, initialized population is divided into $G$ subgroups, objective function $f$ and period of $R$ for executing the communication strategy that the bats are assigned to. For the second step, the compacted subsets are evaluated, the communication scheme activates, and assessed results are compared to find the current best solution. For the third step, the termination is checked for the terminating condition, go to the second step if the termination condition is not met, otherwise it records the best bats and obtains the value of the function $f(x)$.

Scheme 7 shows the overall pseudo code for pcBA, in which $G$ is subpopulations; $n$ is the number of bats in each group; $m$ is the number of groups; $R$ is the exchanging period, and *cBA* is a compact scheme.

---

**Scheme 7** Pseudo code for Parallel and Compact Bats Algorithm-*pcBA*

---

　1:　　*Step 1.* Initialization
　2:　　　generate $G_{1...m}$ $\left(m \leq N_p\right)$ subgroups, each $G$ has $n$ bats
　3:　　　　assign period exchanging time $R$, counter $t = 1$
　4:　　　　solutions $x_{ij}^t$ in the *j-th* subgroup with $n_j$ bats, $i = 1,2, \ldots ,n$; $j = 1,2, \ldots m$
　5:　　*Step 2.* **while** *termination is not satisfied* **do**
　6:　　　　**for** $j = 1$ to $m$ **do**
　7:　　　　　*cBA($G_i$)* according to Scheme 6
　8:　　　　　**end do**
　9:　　　　**if**(*mod(t,R)==0*) **then**
　10:　　　　*Communication ($G_{1...m}$)*; according to Scheme 1
　11:　　　　Find the current best solution $x_{best}$
　12:　　　　$t = t + 1$
　13:　　　　**end while**
　14:　　*Step 3.* Output the best solutions found

---

## 4. Experiment with Numerical Optimization Problems

To evaluate the performance of the proposed pcBA, fifteen optimal numerical problems selected as benchmark functions [33]. Table 1 lists the initialized range, number of variables as the dimension of the space search, and the max iteration for fifteen test functions. The experimental results of the proposed pcBA are compared with the various version of other algorithms of BA, e.g., BA [4], parallel BA (pBA) [23], and compact BA (cBA) [7], as shown in Table 2. Table 3 depicts the obtained results of the proposed pcBA compared with popular metaheuristic algorithms in the literature, e.g., Particle swarm optimization (PSO) [34], Differential evolution (DE) [35], Grey wolf optimizer (GWO) [36], and Genetic algorithm (GA) [25]. Table 4 displays the comparison of the proposed algorithm with four other compact algorithms in the literature, e.g., rcGA [37], cDE [38], cABC [39] and cFPA [40] regarding solution quality and time running. The obtained results of minimized outcomes are averaged for sequences of each testing function with the initialized range, the dimension, and max iteration in Table 1.

**Table 1.** Fifteen selected benchmark functions.

| Name | Test Functions | Range | Dimension | Iteration |
|------|----------------|-------|-----------|-----------|
| Rosenbrock | $f_1(x) = \sum\limits_{i=1}^{n-1}\left(100 \times \left(x_{i-1} - x_i^2\right)^2 + (1 - x_i)^2\right)$ | ±100 | 30 | 2000 |
| Quadric | $f_2(x) = \sum\limits_{i=1}^{n}\sum\limits_{k=1}^{i}(x_i)$ | ±100 | 30 | 2000 |
| Ackley | $f_3(x) = 20 + e - 20e^{-0.2\sqrt{\frac{\sum_{i=1}^{n}x_i^2}{n}}} - e^{\frac{\sum_{j=1}^{n}\cos(2\pi x_i)}{n}}$ | ±32 | 30 | 2000 |
| Rastrigin | $f_4(x) = \sum\limits_{i=1}^{N}[10 + x_i^2 - 10\cos 2\pi x_i]$ | ±5.12 | 30 | 2000 |
| Griewangk | $f_5(x) = 1 + \sum\limits_{i=1}^{N}\frac{x_i^2}{4000} + \prod\limits_{i=1}^{N}\cos\frac{x_i}{\sqrt{i}}$ | ±100 | 30 | 2000 |
| Spherical | $f_6(x) = \sum\limits_{i=1}^{N}\left(x_i^2\right)$ | ±100 | 30 | 2000 |
| Quartic Noisy | $f_7(x) = random[0,1) + \sum\limits_{i=1}^{N}\left(i \times x_i^4\right)$ | ±1.28 | 30 | 2000 |
| Schwefel | $f_8(x) = 418.983n - \sum\limits_{i=1}^{N}x_i \times \sin\left(\sqrt{|x_i|}\right)$ | ±100 | 30 | 2000 |
| Langermann | $f_9(x) = \sum\limits_{i=1}^{n}\left[x_i^2 - 10\cos(2\pi x_i) + 10\right]$ | ±5.12 | 30 | 2000 |
| Shubert | $f_{10}(x) = -20\exp\left(-0.2\sqrt{\frac{1}{n}\sum\limits_{i=1}^{n}x_i^2}\right) -$ $\exp\left(\frac{1}{n}\sum\limits_{i=1}^{n}\cos(2\pi x_1)\right) + 20 + e$ | ±32 | 30 | 2000 |
| Dixon & Price | $f_{11} = (x_1 - 1)^2 + \sum\limits_{i=2}^{d}i\left(2 \times x_i^2 - x_{i-1}\right)^2$ | ±32 | 30 | 2000 |
| Michalewicz | $f_{12} = -\sum\limits_{i=1}^{d}\sin(x_i) \times \sin^{20}\left(\frac{i \times x_i^2}{\pi}\right)$ | ±5.12 | 30 | 2000 |
| Schaffer N.2 | $f_{13} = \frac{1}{2} + \frac{\sin^2\left(x_1^2 - x_2^2\right) - 0.5}{\left[1 + 0.001 \times \left(x_1^2 - x_2^2\right)\right]^2}$ | ±100 | 30 | 2000 |
| Matyas | $f_{14} = 0.26\left(x_1^2 + x_2^2\right) - 0.48x_1x_2 - 10$ | ± 10 | 30 | 2000 |
| Drop-Wave | $f_{15} = \frac{1 + \cos\left(12\sqrt{x_1^2 + x_2^2}\right)}{0.5\left(x_1^2 + x_2^2\right) + 2}$ | ±5.12 | 30 | 2000 |

**Table 2.** Comparison of the proposed pcBA with the BA cBA, and pBA algorithms.

| Functions | pcBA | BA | r | cBA | r | pBA | r |
|---|---|---|---|---|---|---|---|
| $f_1(x)$ | **9.20E−01** | 1.56E+00 | + | 1.07E+00 | + | 9.56E−01 | ~ |
| $f_2(x)$ | **3.38E+00** | 4.34E+00 | + | 3.68E+00 | ~ | 3.49E+00 | + |
| $f_3(x)$ | **1.53E+00** | 4.11E+00 | + | 2.10E+00 | + | 1.96E+00 | + |
| $f_4(x)$ | 4.29E−01 | 5.62E−01 | + | 4.55E−01 | + | 4.29E−01 | ~ |
| $f_5(x)$ | **1.17E+01** | 2.28E+01 | + | 2.44E+01 | + | 1.19E+01 | ~ |
| $f_6(x)$ | **2.15E+00** | 6.79E+00 | + | **2.15E+00** | ~ | 2.38E+00 | + |
| $f_7(x)$ | **2.64E+00** | 4.13E+00 | + | 3.61E+00 | + | **2.46E+00** | - |
| $f_8(x)$ | **−4.37E+02** | −3.67E+02 | ~ | −4.09E+02 | ~ | −4.17E+02 | ~ |
| $f_9(x)$ | **8.57E+01** | 1.38E+02 | + | 1.15E+02 | + | 9.57E+01 | + |
| $f_{10}(x)$ | 1.93E+00 | 1.93E+00 | - | 1.96E+00 | ~ | **1.70E+00** | - |
| $f_{11}(x)$ | **4.70E−02** | 1.34E−01 | + | 6.06E−02 | ~ | 4.16E−01 | + |
| $f_{12}(x)$ | **1.91E−01** | 5.48E−01 | + | 2.83E−01 | + | 6.38E−01 | + |
| $f_{13}(x)$ | 2.08E+00 | 3.17E+00 | + | 2.29E+00 | + | 2.63E+00 | + |
| $f_{14}(x)$ | 9.79E+00 | 1.14E+01 | + | **8.86E+00** | - | **8.14E+00** | - |
| $f_{15}(x)$ | **9.82E−03** | 2.78E−02 | ~ | 1.57E−02 | + | 3.36E−02 | ~ |
| AVG | −2.10E+01 | −1.12E+01 | + | −1.03E+01 | + | −1.90E+01 | + |
| **Summary** | | 13+ 2~ 1- | | 10+ 5~ 1- | | 8+ 5~ 3- | |

**Table 3.** Comparison of the proposed pcBA with the DE, PSO, GWO, and GA methods.

| Functions | pcBA | DE | r | PSO | r | GWO | r | GA | r |
|---|---|---|---|---|---|---|---|---|---|
| $f_1(x)$ | 9.20E−01 | 9.31E−01 | ~ | **7.54E−01** | - | 1.09E+00 | + | 1.25E+00 | + |
| $f_2(x)$ | 3.38E+00 | 3.34E+00 | + | 3.49E+00 | + | 4.09E+00 | + | 4.38E+00 | + |
| $f_3(x)$ | 1.45E+00 | **1.24E+00** | + | 1.91E+00 | + | 2.19E+00 | + | 2.91E+00 | + |
| $f_4(x)$ | **4.29E−01** | 5.93E−01 | + | 4.37E−01 | + | 4.93E−01 | ~ | **4.60E−01** | ~ |
| $f_5(x)$ | **1.17E+01** | 1.21E+01 | + | **1.04E+01** | - | 1.34E+01 | + | 1.98E+01 | + |
| $f_6(x)$ | **2.15E+00** | 2.20E+00 | ~ | 2.38E+00 | + | 2.40E+00 | ~ | 2.17E+00 | ~ |
| $f_7(x)$ | **2.64E+00** | 2.92E+00 | + | 2.46E+00 | ~ | 3.12E+00 | + | 7.40E+00 | + |
| $f_8(x)$ | **−4.33E+01** | −3.63E+01 | + | −2.70E+01 | + | −8.46E+00 | + | −1.25E+01 | + |
| $f_9(x)$ | **5.76E+00** | 5.39E+00 | ~ | 6.40E+00 | + | 6.39E+00 | + | 6.80E+00 | + |
| $f_{10}(x)$ | 1.90E+00 | **2.40E+00** | - | 2.69E+00 | + | 2.36E+00 | + | 2.19E+00 | + |
| $f_{11}(x)$ | **4.70E−02** | 4.16E−01 | + | **4.16E−01** | + | 4.16E−01 | + | 1.22E+00 | + |
| $f_{12}(x)$ | **1.92E−01** | 3.81E−01 | + | **2.38E−01** | ~ | 3.90E−01 | + | 3.75E−01 | + |
| $f_{13}(x)$ | 2.08E+00 | 2.63E+00 | + | 2.63E+00 | + | 2.63E+00 | + | 3.95E+00 | + |
| $f_{14}(x)$ | 9.79E+00 | 1.03E+01 | ~ | 1.11E+01 | + | 9.30E+00 | - | 1.27E+01 | + |
| $f_{15}(x)$ | **9.82E−03** | 8.33E−02 | + | 3.36E−01 | ~ | 8.33E−02 | + | 6.47E−02 | ~ |
| AVG | −5.88E−02 | 5.73E−01 | + | 1.24E+00 | + | 2.66E+00 | + | 3.55E+00 | + |
| **Summary** | | 11+ 4~ 1- | | 11+ 3~ 2- | | 12+ 3~ 1- | | 13+ 3~ 0- | |

Parameters setting for the algorithms occurs as follows. Virtual and real population size *N* of the mentioned algorithms set to 80. The dimension of the solutions space-*D* is set based on the problem dimension requirements listed in Table 1. Full iterations for each function are set to 2000. Several runs for each testing function are set to 25. Some of the subgroups *m* are set to 2, 4, and 8. An exchanging period *R* is established in a loop of 20 times current iterations. The further setting is referenced in reference [6]. The final results are taking average of the outcomes from all runs.

The compared results of the proposed pcBA with various bats algorithms, e.g., the BA, cBA, and pBA, are shown in Table 2. A parameter r is a ratio that is a paired comparison between pcBA and other algorithms respectively, i.e., pcBA and BA, pcBA and cBA, and pcBA and pBA. The other column values are the mean outcomes of the runs for the functions, respectively. The denoted r is symbols

of '+' '-' and '~' means the 'better' 'worse' and 'approximate' measurements of the deviation with respect to their outcomes, respectively. The highlighted numbers are the best results among them in each function (each row of the table). If the pcBA is better (smaller the value for minimized, or bigger for maximized problems) than the others: BA, cBA, and pBA, then *r* is the symbol '+'. Similarly, the symbols "-" and "~" are for the 'worse' and 'approximated' cases. Visibly, almost all the highlighted cases belong to pcBA for testing the benchmark functions.

Table 3 compares the performance for fifteen numerical optimization problems of the proposed pcBA with the other popular metaheuristic algorithms such as DE, PSO, GWO, and GA. The highlighted numbers are the best results of the obtained average outputs among them in each function.

Table 4 shows the compared performance quality optimization between pcBA and the other compact algorithms such as cABC, cPFA, cDE, and rcGA. The highlighted numbers are the best results of the obtained average outputs among them in each function. As observed in Tables 2–4, the most highlighted number and the symbol "+" of better points belong to the proposed algorithm. That means the proposed approach offers a competitive algorithm.

**Table 4.** Comparison of the proposed pcBA with the cABC, cFPA, cDE, and rcGA respectively for 15 test functions.

| Functions | pcBA | cABC | r | cFPA | r | cDE | r | rcGA | r |
|---|---|---|---|---|---|---|---|---|---|
| $f_1(x)$ | **9.25E−01** | 9.54E−01 | ~ | 7.39E+00 | + | 1.51E+00 | + | 9.44E−01 | ~ |
| $f_2(x)$ | 3.38E+00 | **2.99E+00** | - | 1.15E+01 | + | 4.61E+00 | + | 1.02E+01 | + |
| $f_3(x)$ | **1.53E+00** | 1.51E+00 | ~ | 6.10E+00 | + | 4.92E+00 | + | 7.45E+00 | + |
| $f_4(x)$ | 3.67E−01 | **1.53E−01** | - | 7.94E−01 | + | 6.98E−01 | + | 7.99E−01 | + |
| $f_5(x)$ | **1.17E+01** | 1.34E+01 | + | 1.11E+01 | ~ | 2.23E+01 | + | 1.76E+01 | + |
| $f_6(x)$ | 2.15E+00 | 3.91E+00 | + | 1.01E+01 | + | **1.72E+00** | - | 5.17E+00 | + |
| $f_7(x)$ | **2.64E+00** | 8.81E+00 | + | 2.05E+01 | + | 5.49E+00 | + | 7.40E+00 | + |
| $f_8(x)$ | **−4.37E+01** | −2.50E+01 | + | −2.67E+01 | ~ | −2.38E+01 | + | −1.25E+01 | + |
| $f_9(x)$ | **8.57E+01** | 1.21E+02 | + | 1.08E+02 | + | 7.94E+01 | - | 1.28E+02 | + |
| $f_{10}(x)$ | 1.93E+00 | **1.66E+00** | - | 3.55E+00 | + | 1.81E+00 | + | 3.18E+00 | + |
| $f_{11}(x)$ | **4.70E−02** | 5.93E−02 | ~ | 3.46E−01 | + | 9.39E−02 | ~ | 3.22E−01 | + |
| $f_{12}(x)$ | **4.91E−01** | 9.81E−01 | + | 1.75E+00 | + | 8.24E−01 | + | 1.32E+00 | + |
| $f_{13}(x)$ | 2.08E+00 | 6.93E−01 | - | 4.67E+00 | + | **6.07E−01** | - | 1.95E+00 | - |
| $f_{14}(x)$ | 9.79E+00 | 1.22E+01 | + | 1.01E+01 | + | 1.27E+01 | + | 1.23E+01 | + |
| $f_{15}(x)$ | **9.82E−03** | 2.40E−02 | ~ | 1.93E+00 | + | 6.32E−02 | ~ | 2.75E−02 | ~ |
| AVG | 5.27E+00 | 9.55E+00 | + | 1.14E+01 | + | 7.52E+00 | + | 1.23E+01 | + |
| **Summary** | | 8+<br>4~<br>4- | | 14+<br>2~<br>0- | | 11+<br>2~<br>3- | | 13+<br>2~<br>1- | |

Figure 2 illustrates the comparison of executing time of the proposed pcBA with the BA, PSO, GWO, cBA, and pBA algorithms for the first eight functions. Clearly, all cases of the time consumption for testing functions of the pcBA are smaller than the other algorithms, but the shortest running time belongs to cBA. The results of the fast processing speed are that some memory-stored parameters of cBA and pcBA are smaller than the stored solutions in the population-based algorithms.

Figures 3–5 show the compared the best score results of the proposed pcBA with rcGA, cDE, cABC, cFPA, and cBA for three selected testing functions $f_7(x)$, $f_8(x)$ and $f_9(x)$ over 25 runs outputs in the same iteration of 2000. Clearly, the cases of these testing functions on the pcBA (indicated red curve) shows a comparatively faster convergence than other algorithms. It says the accuracy of the proposed pcBA is improved significantly.

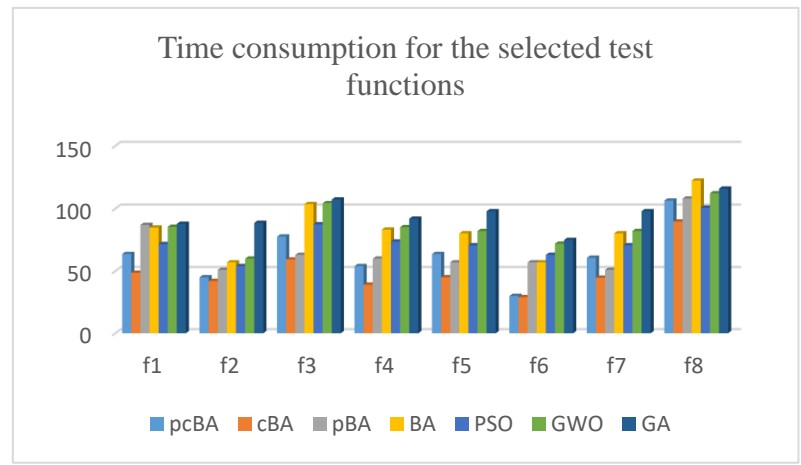

**Figure 2.** Comparison of running times of the pcBA, with the cBA, pBA, BA, PSO, GWO, and GA algorithms for the first eight test functions.

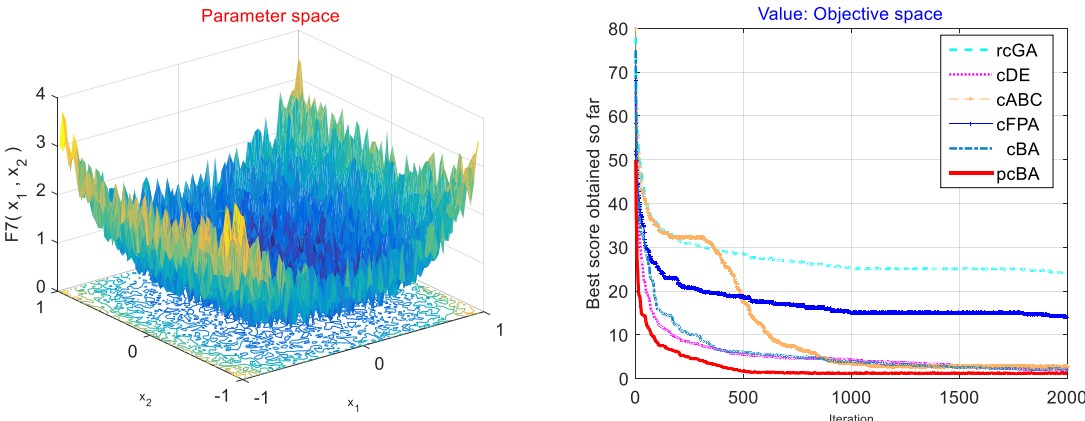

**Figure 3.** Comparison of the evaluated performance of the proposed pcBA with the rcGA, cDE, cABC, cFPA, and cBA algorithms for the testing function f7 of Quartic Noisy.

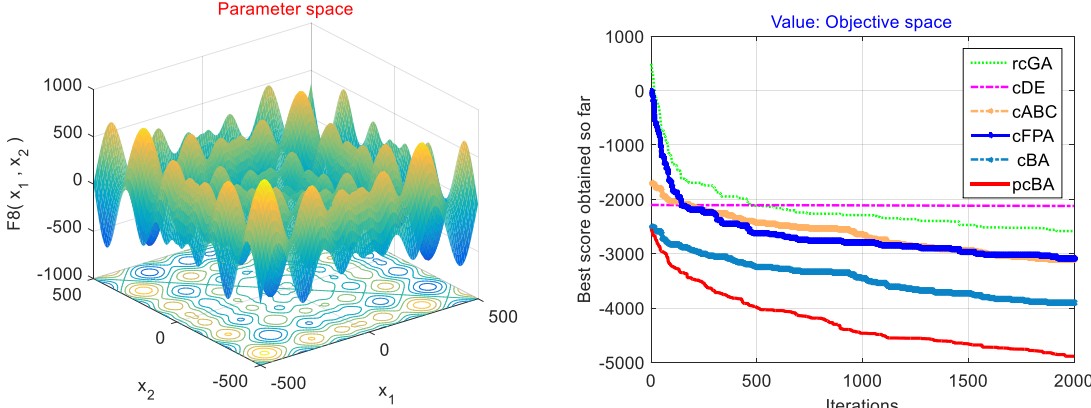

**Figure 4.** Comparison of performance of the proposed pcBA with rcGA, cDE, cABC, cFPA, and cBA algorithms for the testing function f8 of Schwefel.

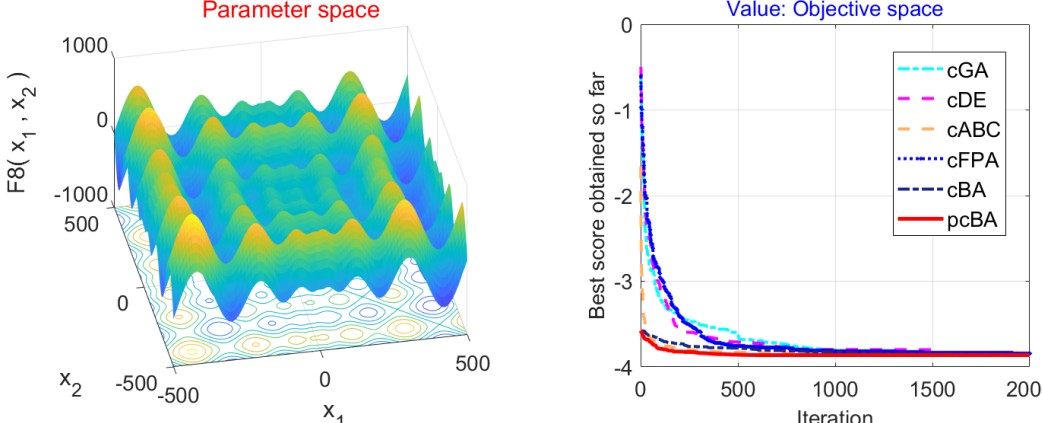

**Figure 5.** Comparison of performance of the proposed pcBA with rcGA, cDE, cABC, cFPA, and cBA algorithms for the testing function f9 of Langermann.

## 5. Applied pcBA for Optimally Balanced Energy Consumption

The unbalanced energy consumption in WSN with multihop communications is one that causes the hotspot problem. The energy consumption of CH nodes closer to BS is increased more than the others because of the massive traffic flows. In this section, we apply the pcBA for optimally balanced energy consumption in deployment WSN. We utilize adjustable parameters of loudness of the bats in BA and a distance coefficient ratio in the objective function for dealing with the mentioned hotspot issue. The loudness $A_i$ of BA can vary for responding to distances in clustering criterion for unequal clustering in WSN. The optimized total communication distances from the cluster members to CHs, and CHs to BS in WSNs provide the saving energy increase. The sequence of experiments consists of steps: modeling the objective function, describing proper agent representation, setting up a mapping solution model, and comparing results.

### 5.1. Objective Function

In designing and deploying sensor networks, a prolonging the lifetime is a core demand. A crucial factor in extending the WSN lifetime is to reduce the energy consumption of its entire network. The power consumption of WSNs is affected directly by the clustering criterion problem. The employed heuristic clustering approaches by evaluating the fitness function is one of the most efficient ways to deal with this issue. The objective function has also evaluated the WSN performance. This objective function consists of the mean consumed energy for round $t$ in Equation (8) and the standard deviation of residual energy in Equation (11). Each sensor node member is connected to the one closest CH after the CHs are decided. Equation (1) is used to get $E_t(n)_{consumed}$ with CH node ($i \in CH$, $x_i = 1$) or the non-cluster head node ($i \in non-CH$, $x_i = 0$).

$$Minimize\ F(x) = \omega \times \mu(E_{consumed}) + (1-\omega) \times \delta(E_{res.}) \tag{18}$$

where $\omega$ is the weight of average consumed energy and standard deviation of residual energy. Table 5 tabulates an example of the residual energy of round 11–12, and the power consumption of round 11 for the ten node network example. Assuming we get node 4, with the $\mu(E_{consumed}) = 0.000204$, $\mu(E_{res.}) = 0.4986160$, $\delta(E_{res.}) = 0.000237$, the obtained result of the applied function in Equation (18) is 0.0002205 (with $\omega$ is set to 0.5) for the ten nodes network.

**Table 5.** An example of the residual and consumed energy of 11–12 round of the ten nodes network.

| Node $i$ | $E_{11(n)_{res.}}$ | $E_{11(n)_{consued}}$ | $E_{12(n)_{res.}}$ |
|:---:|:---:|:---:|:---:|
| 1 | 0.4984650 | 0.000090 | 0.4983750 |
| 2 | 0.4985640 | 0.000056 | 0.4985080 |
| 3 | 0.4984960 | 0.000070 | 0.4984260 |
| 4 | 0.4986160 | 0.000204 | 0.4984120 |
| 5 | 0.4985520 | 0.000077 | 0.4984750 |
| 6 | 0.4985480 | 0.000079 | 0.4984690 |
| 7 | 0.4984300 | 0.000364 | 0.4980660 |
| 8 | 0.4977127 | 0.000068 | 0.4976447 |
| 9 | 0.4985010 | 0.000074 | 0.4984270 |
| 10 | 0.4987388 | 0.000065 | 0.4986738 |

## 5.2. Balancing Load Clusters

The hot spot problem in WSN prevents from figuring out the balancing load based on the layout of uneven clustering. The partitioned clusters in the network have different sizes. The closer clusters to BS are the hotter cluster because the traffic relay load of the CHs near BS that suffered heavier than those CHs are farther away from BS. To avoid this problem, we figure out the adjustable parameters are not only related to the objective function but are also related to the optimization algorithm. Two possible coefficients need to be adjusted: the first is changing the distance applied to the objective function, and the second is adjusting the loudness of the optimization algorithm. For the objective function of constructing unequal clusters, each CH needs to adjust its equal cluster distances. Let's $R_c$ be the distance adjustment factor, used as follows:

$$d_{CH_j} = d_{CH_j} \times R_c \tag{19}$$

where $d_{CH_j}$ is the distance from node $CH_j$ to the BS, and $R_c$ is a ratio adjusting parameter. This $R_c$ is calculated as:

$$R_c = \left[ 1 - \alpha \frac{d_{max} - d_{CH_j}}{d_{max} - d_{min}} - \beta \left( 1 - \frac{E_{res.}}{E_{max}} \right) \right] \times R_{max} \tag{20}$$

where $d_{max}$ and $d_{min}$ are the maximum and minimum distance from the CHs in the network to the BS; $R_{max}$ is the maximum value of competition radius; $\alpha$ is a weighted factor whose value is in [0, 1]; $E_{res.}$ is the residual energy of $CH_j$. Balancing the load of WSN based on clustering formation by updating Equation (7), and performing the optimal objective function by Equation (18) with the assistance of different distances Equations (19) and (20). Figure 6a depts the comparison of the applied pcBA for balancing energy consumption in WSNs with the BA, PSOTVW, and PSOTVAC approaches. It is seen that the applied pcBA outstands performance the other approaches in terms of coverage rate. Additionally, an adjusted distance of CHs to BS is applied to the pcBA for the objective function that obtained the better than nonadjusted length, as shown in Figure 6b.

For the optimization algorithm, we assigned the variable of loudness $A_i$ of BA as in Equation (5) to correspond to the radius of the changing cluster size by bias iteration. We expressed the loudness of BA mathematically the time-varying through the following equation:

$$A_i^0 = (A_{max} - A_{min}) \times \frac{(MaxIteration - Iter)}{MaxIteration} + A_{min} \tag{21}$$

where *MaxIteration* is number of the maximum iteration, *Iter* is the current time steps, and $A_{max}$, $A_{min}$ are constants set to 0.5 and 0.25, respectively. The time varying loudness is mathematically represented in Equation (21) of pcBA for the hot spot problem in WSN. Figure 7a shows the comparison of some nodes alive for WSNs of advanced pcBA with PSOTVAC, PSOTVIW-WSN, and LEACH

methods; and Figure 7b depicts the adjusted pcBA-WSN with unadjusted pcBA-WSN for a different loudness parameter.

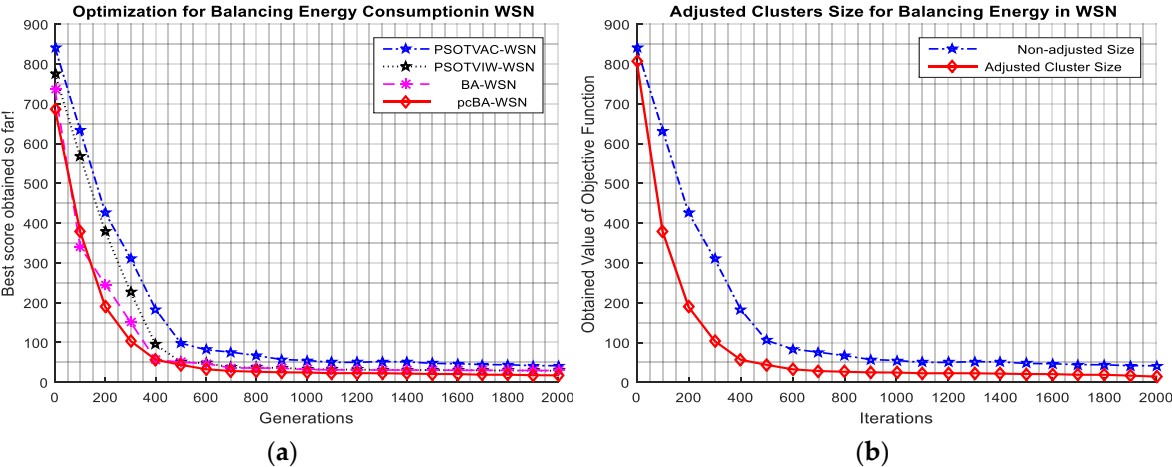

(**a**)                                                                                           (**b**)

**Figure 6.** Comparison of the advanced pcBA with others for balancing energy consumption in WSNs. (**a**) Comparison of average outcomes of 25 runs of four approaches of applied pcBA, BA, two cases of PSO for minimizing objective function Equation (18); (**b**) Applied pcBA for adjusted distances of CHs to BS.

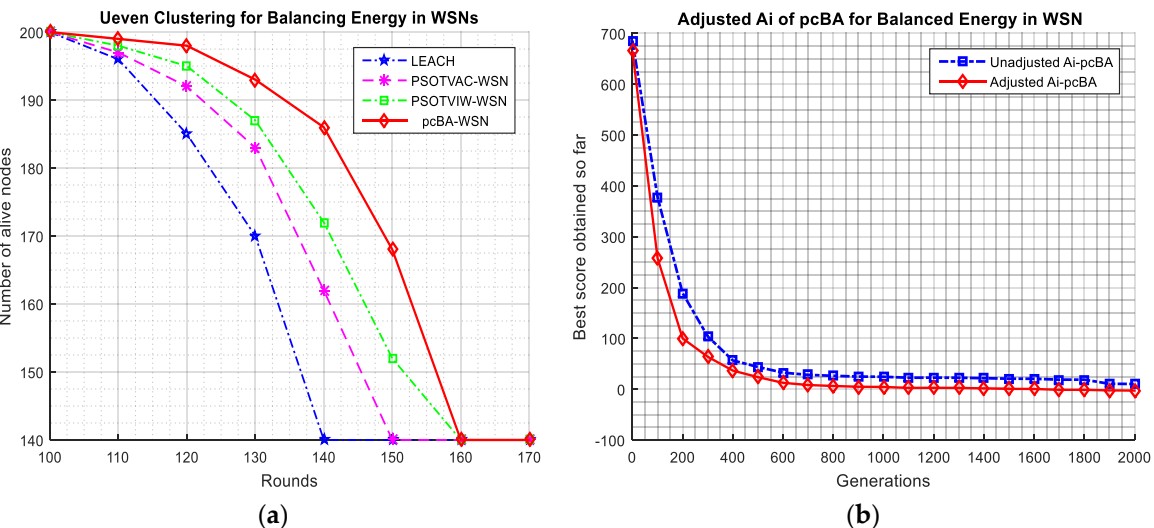

(**a**)                                                                                           (**b**)

**Figure 7.** Comparison of adjusted pcBA-WSN with unadjusted pcBA-WSN for a varying loudness parameter. (**a**) Comparison of some nodes alive for WSNs of advanced pcBA with PSOTVAC, PSOTVIW-WSN, and LEACH methods; (**b**) Comparison of adjusted pcBA-WSN with regular pcBA-WSN.

### 5.3. Structured Model Solution

The model solution is structured as follows: a two-dimensional (x-y axes) coordinate system, generating nodes randomly with its corresponding coordinates, initial clustering with the binary CHs property, calculating distances nodes to CH, CHs to BS, and finally optimizing load balancing by applying pcBA. Table 6 shows an example of the position of nodes in a network area. A modeled WSN is a graphs G with $N$ nodes distributed randomly in desired areas. A simulated network with $N$-nodes ($N = 100, 200 \ldots$ ) is also distributed in a 2-D space $[0{:}M, 0{:}M]$ ($M = 100, 200 \ldots$ ).

**Table 6.** A sample of expressing the positions of the sensor nodes.

| Index | Node$_i$ | 1 | 2 | 3 | 4 | 5 | 6 | 7 | .. | N |
|---|---|---|---|---|---|---|---|---|---|---|
| x | 05 | 65 | 95 | 100 | 75 | 60 | 45 | 40 | .. | 10 |
| y | 01 | 5 | 10 | 30 | 15 | 20 | 45 | 60 | .. | 80 |

Each node can communicate with others by using *r* transmission range. Node *i* can receive the signal of node *j* if node *i* is in the transmission range *r* of node *j*.

Table 7 indicates the attribution of existing CHs in the WSN. This is the integer model with a binary decision variable if $CH_j = 1$, it is the selected CH in WSN, and $CH_j = 0$, otherwise it is the not selected CH. The initial values of communication energy parameters is referred to in reference [41].

**Table 7.** The attribution of existing cluster head (CHs) if flag = 1, Node is set to CH, otherwise Node is configured to member node.

| Index | Node$_i$ | 1 | 2 | 3 | 4 | 5 | 6 | 7 | 8 | .. | n |
|---|---|---|---|---|---|---|---|---|---|---|---|
| CH | *i* | 0 | 0 | 0 | 0 | 1 | 0 | 0 | 0 | .. | 0 |

In the target network, there are *N* deployed nodes in a $n \times n$ grid space where a test platform is established, where nodes were randomly distributed between (*x* = 0, *y* = 0) and (*x* = *n*, *y* = *n*) with *N* set to 100, 200, 300 and 400 node networks. The objective function is in Equation (18) to be repeated in generations of 2000 by different random seeds with 25 runs. Table 8 displays the initial values of the parameters for setting the experiment for optimally balancing energy consumption.

**Table 8.** Initial values of parameters for setting the experiment of optimally balancing energy consumption.

| Parameters Noticed | Denoted Symbols | Initial Values |
|---|---|---|
| Initial node energy | $E_j$ | 0.5 J |
| Data aggregation energy | $E_{DA}$ | 5 nJ/bit/signal |
| Receiving and transmitting energy | $E_{fs}$ | 10 pJ/bit/m$^2$ |
| Radio electronics energy | $E_{elec}$ | 50 nJ/bit |
| Number bit of a data message | *l* | 1024 bit |
| Amplifier energy | $E_{mp}$ | 0.013 pJ/bit/m$^4$ |
| Number of nodes in WSN | N | 100/200/300/nodes |
| Space distribution | M | 100/200/300 m |
| Population size (or virtual size for compact) | *Pop* | 40 |
| Iteration (generations) | *MaxIteration* | 2000 |
| Maximum of the loudness of BA | $A_{max}$ | 0.5 |
| Minimum of the loudness of BA | $A_{min}$ | 0.25 |
| Minimum bats' frequency | $f_{min}$ | 0 |
| Maximum bats' frequency | $f_{max}$ | Number of nodes |
| Bats' pulse emission | *r* | 0.35 |
| Number of runs | *runs* | 25 |
| Exchanging time for communication | R | 25 |

Figure 8 shows the detailed steps of processing in applied pcBA for balancing load in WSN.

Step 1: Initialize parameters: subgroups G, pulse rate $r_i$ , the loudness $A_i$, $\beta$ are set to *random;* define pulse frequency $f_{min}$, $f_{max}$ as search range.

Step 2: Assign $F_{min}$ to fitness($x_{best}$), with $x_{best}$- the best solution by means of perturbation vector PV; Generate solution $x^t$ from $PV$; assign exchanging time $R$, counter $t = 1$

Step 3: Update velocities $v^t$ and locations $x^t$ according to updating rule of BA; if it is selected best solution; assign the function value $F_{new}$ to fitness($x^{t+1}$).

Step 4: Compare $x^{t+1}$ and $x_{best}$, let be the one with *winner*, and the other one is *loser*.

Step 5: Update PV, *if (mod(t,R)==0 ) then Communication* ($G_{1..m}$); Find the current best solution $x_{best}$

Step 6: Accept a new solution if the solution improves ($F_{new}$ less than $F_{min}$), then updating global best $x_{best}$ to $x^{t+1}$ and assign function $F_{min}$to $F_{new}$

Step 7: If it is not meet the termination condition go to step 3.

Step 8: Output the global best solution $x_{best}$

**Figure 8.** The steps of an applied pcBA for balancing load in WSN.

*5.4. Experimental Results*

We can figure out the optimization problem by minimizing the objective function in Equation (18). The experimental results of the applied pcBA are compared with two cases for PSO: the time-varying inertia weight (PSO-TVIW), and time-varying acceleration coefficients (PSO-TVAC) [21] and the BA [42] by regarding solution quality and speed. The obtained result is averaged from the outcomes from all runs. Figure 6a indicates four of the curves of the pcBA – WSN, BA – WSN, PSOTVIW – WSN and PSOTVAC – WSN methods for minimizing the objective function in Equation (18). Apparently, pcBA-WSN is as good as BA-WSN and faster than the PSOTVIW-WSN and PSOTVAC-WSN approaches regarding convergence. With the support of adjustable parameters in Equations (19)–(21), the size of clusters is varied based on responding to these variables to avoid the unbalanced load problem in WSNs. Figures 6 and 7 show the results of the efficiently adjusting parameters. Figure 7 compares the performance qualities in two cases of the selected parameter of loudness of applied pcBA for adjusted cluster size with none of the selected altered loudness setting. Visibly, the average fitness values of the pcBA method in the case of adjusting $A_i$ for optimal WSN are better than the regular case of convergence for removing the hotspot problem.

Table 9 compares the performance quality and running time for optimizing the objective function Equation (18) in WSNs of four methods of the pcBA – WSN, PSOTVAC – WSN, PSOTVIW – WSN, and the BA – WSN. Clearly, the average value of pcBA for the objective function is faster than those obtained by other approaches of PSO. The applied pcBA for optimizing the clustering WSN is not as different as convergence much of using BA. However, the total time consumption of pcBA method is fastest at only 2.045 min due to working memory variables being smaller. The obtained results are the average of the outcomes from all runs and are compared with a variety of versions of BA, and others in the literature.



**Table 9.** The comparison of outcomes and computation time of pcBA-WSN with other methods, e.g., BA-WSN, PSO-TVAC, and PSO-TVIW for minimizing objective function.

| Methods | Pop. Size | Iterations/a run | Min | Max | Mean | Std. | Running Time (m) |
|---|---|---|---|---|---|---|---|
| PSO – TVAC [21] | 40 | 2000 | 5.72E+01 | 8.41E+02 | 2.57E+02 | 2.87E+02 | 3.01E+00 |
| PSO – TVIW [21] | 40 | 2000 | 6.80E+01 | 6.35E+02 | 2.25E+02 | 2.83E+02 | 3.10E+00 |
| BA-WSN [42] | 40 | 2000 | 6.93E+01 | 7.57E+02 | 2.03E+02 | 2.69E+02 | 3.26E+00 |
| **The applied pcBA-WSN** | **1x4** | **2000** | **5.65E+01** | **7.65E+02** | **2.01E+02** | **2.32E+02** | **2.04E+00** |

In other experiments, the performances of applied pcBA can be compared to previous methods (LEACH, LEACH-C [24,41], and HEED [43]), as illustrated in Table 10, in which FND and LND are the first nodes die, and the last nodes die respectively. Apparently, the overall performance of the applied pcBA results in a longer lifetime of the nodes than other methods.

**Table 10.** Comparisons of the obtained average of applied pcBA with using other methods for a case of N = 100 nodes of WSN.

| Number of Nodes | Methods | The Round at FND | The Round at LND | Total SMS Packages |
|---|---|---|---|---|
| 100 (0, 0) | **Applied pcBA** | **4328** | **4446** | **439389** |
| | HEED [43] | 3684 | 4432 | 432564 |
| | LEACH-C [41] | 4140 | 4272 | 414937 |
| | LEACH [24] | 3504 | 3902 | 383441 |
| 100 (center) | **Applied pcBA** | **4612** | **6627** | **654988** |
| | HEED [43] | 4612 | 6798 | 669816 |
| | LEACH-C [41] | 4308 | 4333 | 428438 |
| | LEACH [24] | 3586 | 4182 | 404223 |

The energy consumption in the network is the majority of CHs. Figure 9 compares the average residual energy of performance measures for a case of a 100 nodes system of LEACH, LEACH-C, HEED, and the related pcBA methods. Obviously, the average residual power consumption of applied pcBA optimized is better than those obtained from LEACH, LEACH-C, HEED, for this network.

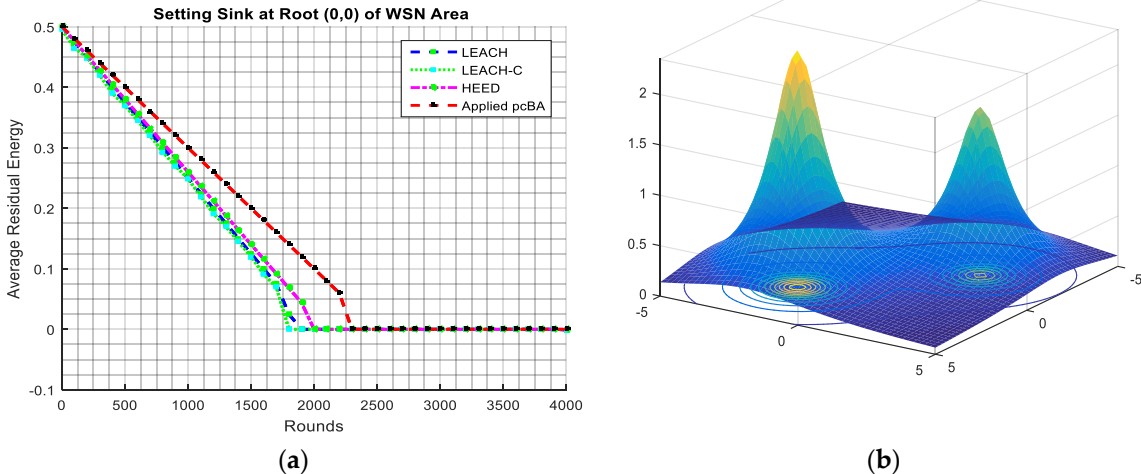

(a)                                                                          (b)

**Figure 9.** Comparison of consumed average residual energy of the applied pcBA for WSN with LEACH, LEACH-C, HEED methods. (**a**) Comparison of consumed average residual energy of the applied pcBA for WSN with LEACH, LEACH-C, HEED methods; (**b**) The space objective function.

Figure 10 shows the applied pcBA performance for 200 nodes in WSN regarding some nodes alive in comparison with those obtained from LEACH, LEACH-C, HEED methods. Obviously, the figure of the proposed pcBA is better than those obtained from LEACH, LEACH-C in both cases of Sink at the root (0, 0) and at the center.

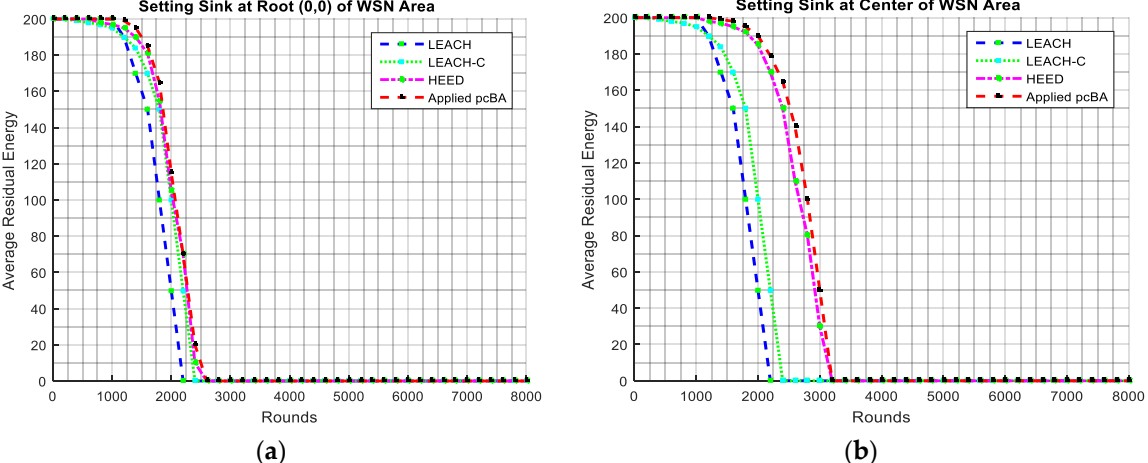

**Figure 10.** Comparison of the number of nodes alive for a configured network with 200 nodes of the applied pcBA with LEACH, LEACH-C, HEED methods. (**a**) Comparison of the number of nodes alive for 200 network nodes of the applied pcBA with LEACH, LEACH-C, HEED methods with Sink at the root (0, 0); (**b**) Comparison the number of nodes alive for 200 network nodes of the applied pcBA with LEACH, LEACH-C, HEED methods with Sink at the center ($x_{max}/2$, $y_{max}/2$).

## 6. Conclusions

In this paper, we proposed pcBA, a new optimal method based on a hybrid of the parallel and compact techniques for the Bats algorithm (BA) for optimization problems and applied it to an energy balance problem in Wireless sensor networks (WSN). The implementation of hybrid technology shows significant advantages from each of the test algorithms and achieves improved collaboration in the optimization algorithm. These methods avoid the optimum local issue in compound constrained optimization problems and allow fast convergence and memory saving. Additionally, we improved the compacting technique by using a controlling weight for adjusting the balance of the probability vector, and we enhanced the parallel techniques by making a communication strategies dynamic. In the simulation section, a set of the selected optimization problems and balanced energy consumption methods in WSNs are used to evaluate the accuracy, executing the time and the saving memory variable of the proposed algorithm. The compared results with BA and the other algorithms in the literature show that the proposed algorithm outperforms its competitors. Also, for balancing the energy consumption problem in WSNs, the result indicates that the proposed approach provides an effective way of utilizing the saving memory variable.

**Author Contributions:** Conceptualization, T.-T.N., and T.-K.D.; Methodology, T.-T.N.; Software, T.-K.D.; Validation, T.-T.N., T.-K.D. and J.-S.P.; Formal Analysis, T.-T.N.; Investigation, J.-S.P.; Resources, J.-S.P.; Data Curation, T.-K.D.; Writing—Original Draft Preparation, J.-S.P.; Writing—Review & Editing, T.-K.D.; Visualization, T.-T.N.; Supervision, J.-S.P.; Project Administration, J.-S.P.; Funding Acquisition, J.-S.P.

**Funding:** This work was supported by the Natural Science Foundation of Fujian Province under Grant No. 2018J01638.

**Conflicts of Interest:** The authors declare no conflict of interest.

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
