# Peer review of "A Novel Improved Bat Algorithm Based on Hybrid Parallel and Compact for Balancing an Energy Consumption Problem"

_information, doi:10.3390/info10060194_

Round 1

Reviewer 1 Report

New method with good results and contribution. Very good paper overall.

Author Response

We would like to sincerely thank Reviewer#1 for careful reading of our manuscript and the constructive comments.

The authors greatly appreciate the positive comments about the topic and applicability of the manuscript. 

The authors checked the entire text for possible grammar and syntax errors. Found errors are corrected, and the changes are marked with red color in the revised manuscript.

A new method with good results and contribution. The very good paper overall.

Thank you again!

Reviewer 2 Report

This paper presents an improved bat algorithm based on
hybrid parallel and compact approaches, then applies
to solve energy consumption problem. There are some
new contributions and the results are valid. So the
paper is recommended to be accepted on the condition
that it is revised carefully.

1) A detailed literature review about the bat algorithm
and balancing energy consumption problem should be provided
so as to put the research into the proper context.

2) Please provide more details about all the key parameter
values used, ideally summarized in a table.

3) There are some typos or grammatical errors.
 a) The title "parellel and compact" seem to a bit odd.
 Please make it clearer.
 b) Eq.(13), there are some typos: the bracket ( ) after erf()
is not closed.  Please check. Also, both "erf" and "exp"
should be in normal font (not italic).

4) Check all references and provide all the details
and make sure they are cited correctly.

Author Response

We would like to sincerely thank Reviewer#2 for careful reading of the manuscript. The authors much appreciate the constructive comment of the Reviewer#2.

In the revised manuscript, we modified section 1 and added more review

We appreciate this suggestion made by Reviewer#2.

In our manuscript's experiment for evaluation of the proposed approach, the selected benchmark functions, and the balancing energy consumption are used for applying the proposed method.

For the selected benchmark functions, some parameters are listed in Table 1; however, for the balancing energy consumption, we have changed the text of initial parameters for setting by summarized in Table 8.

The authors checked the entire text for possible grammar and syntax errors. Found errors are corrected, and the changes are marked with red color in the revised manuscript.

Thank you so much!

Reviewer 3 Report

This is a very interesting work. However you have to made some syntactic corrections. For example:

line 7: "an improved" Bat algorithm

line11: the "selected" benchmark

line 80: in the subgroups "have been replaced"

Author Response

The authors are thankful to Reviewer#3 much for careful reading of the manuscript and meaningful insights on the flaws of our original paper.

 We much appreciate the constructive comment of the Reviewer#3. Found errors are corrected, and the changes are marked with red color in the revised manuscript.

In the revised manuscript, we corrected the typo:

Abstract: This paper proposes an improved Bat algorithm based on hybridizing parallel and compact (namely pcBA) for a class of the saving variables in the optimization problems. The parallel enhances diversity solutions for exploring in space search and sharing computation load. Nevertheless, the compact saves stored variables for computation in the optimization approaches. In the experimental section, the selected benchmark functions, and the energy balance problem in Wireless sensor networks (WSN) are used to evaluate the performance of the proposed method. Compared results with the other methods in the literature demonstrate that the proposed algorithm achieves the practical way of reducing the number of stored memory variables, and running time consumption.

Reviewer 4 Report

This paper proposes an improvement Bat algorithm based on hybridizing parallel and compact (namely pcBA) for a class of the saving variables in the optimization problems. The parallel enhances diversity solutions for exploring in space search and sharing computation load.

Nevertheless, the compact saves stored variables for computation in the optimization approaches. In the experimental section, the select benchmark functions, and the energy balance problem in Wireless sensor networks (WSN) are used to evaluate the performance of the proposed method. Compared results with the other methods in the literature demonstrate that the proposed algorithm achieves the practical way of reducing the number of stored memory variables, and running time consumption.

The authors present a very nice introduction and a very nice definition of the problem and other proposed optimization algorithms reported in the literature. Therefore, it is ready to be accepted in Information Journal.

Author Response

We would like to sincerely thank Reviewer#1 for careful reading of our manuscript and the constructive comments.

The authors much appreciate the positive comments about a very nice introduction and a very helpful definition of the problem and other proposed optimization algorithms reported in the literature.

Thank you again!

Sincerely,
